# Limitations of Neural Collapse for Understanding Generalization in Deep Learning

## Abstract

The recent work of Papyan, Han, and Donoho (2020) presented an intriguing "Neural Collapse" phenomenon, showing a structural property of interpolating classifiers in the late stage of training. This opened a rich area of exploration studying this phenomenon. Our motivation is to study how far understanding Neural Collapse can take us in understanding deep learning. First, we investigate its role in generalization. We refine the Neural Collapse conjecture into two separate conjectures: collapse on the train set (an optimization property) and collapse on the test distribution (a generalization property). We find that while Neural Collapse often occurs on the train set, it does not occur on the test set. We thus conclude that Neural Collapse is primarily an optimization phenomenon, with as-yet-unclear connections to generalization. Second, we investigate the role of Neural Collapse in representation learning. We show simple, realistic experiments where more collapse leads to *worse* last-layer features, as measured by transfer-performance on a downstream task. This suggests that Neural Collapse is not always desirable for representation learning, as previously claimed. Our work thus clarifies the phenomenon of Neural Collapse, via more precise definitions that motivate controlled experiments.

## 1 Introduction

In science, and in deep learning, novel empirical observations often catalyze deeper scientific understanding (Kuhn, 1962). When faced with a new or surprising experiment, we can then try to understand the phenomenon more precisely: How universal is the behavior? In what settings does it hold? Can we describe it quantitatively? What does it teach us more generally? This overall roadmap for understanding —from observations to quantitative conjectures & laws— has a long history of success in the natural sciences, and has also enjoyed recent successes in deep learning.

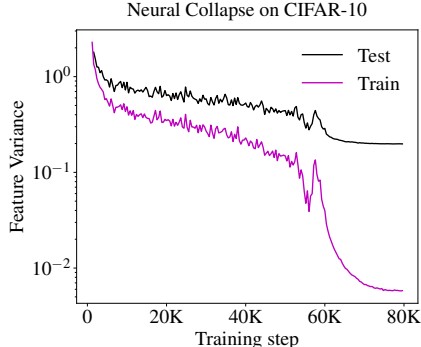

Figure 1: **Failure of Test Collapse.** Neural Collapse for ResNet18 on CIFAR-10. Collapse appears to occurs on the train set, but not on test.

The recent "Neural Collapse" work of Papyan, Han, and Donoho (2020) initiated another instance of such a research program in understanding deep learning. Their work presented a new experimental observation, along with a partial characterization. At a high level, Neural Collapse conjectures several structural properties of deep neural networks when trained past the point of 0 classification error on the train set. Their most relevant conjecture to generalization— is "variability collapse (NC1)." Variability collapse proposes, informally, that when a deep network is trained on a $k$-way classification task, the last-layer representations converge to $k$ discrete points. This is apriori surprising, since this internal structure is in no way required to achieve low train loss and high test performance: there exist networks with identical decision boundaries which do not satisfy collapse. However, our standard training methods (Stochastic Gradient Descent and variants) on standard architectures and datasets empirically seem to satisfy some form of collapse, as demonstrated in Papyan et al.

(2020). This work has since inspired many follow-up works investigating this phenomenon, both theoretically and empirically.

A motivating factor in this research program is the belief that Neural Collapse is not an isolated phenomenon, but rather is deeply connected to other important and unsolved aspects of deep learning— in particular *generalization.* The problem of generalization, informally, is the study of why a model trained on a finite set of samples has good performance on out-of-sample inputs. Although this is not apriori related to Neural Collapse, the original work proposes that collapse "confers important benefits, including better generalization performance, better robustness, and better interpretability." And it is stated as a hypothesis that "the benefits of the interpolatory regime of overparametrized networks are directly related to Neural Collapse" (Papyan et al., 2020). This postulated connection between Neural Collapse and generalization is implicit in many of the follow-up works as well, and motivates studying collapse as a phenomenon.

However, the nature of the connection between Neural Collapse and generalization remains muddled. Some works argue they are closely related (Papyan et al., 2020), while others cast some doubt (Elad et al., 2020; Zhu et al., 2021; Banburski et al., 2021). There are at least two reasons for this confusion in the literature: First, it is often not clear whether Neural Collapse refers to a phenomenon on the train set, or on the test set. The behaviors most relevant to generalization occur on the test set, and yet most experiments and theorems consider only the train set. Second, the Neural Collapse conjectures do not precisely specify the role of the sample size, and thus it is not always clear how to connect to generalization— where sample size is fundamental. This ambiguity is especially problematic because some natural ways to extend the Neural Collapse conjecture to the test set turn out to be impossible to satisfy, as we will describe.

**Our Contributions.** We clarify ambiguities in the original Neural Collapse (NC) conjectures, which allows us to investigate which forms of NC are possible to achieve, both in theory and in practice. Specifically:

1. We propose more precise versions of the Neural Collapse conjectures ("variability collapse"), stating different versions for the train set and the test set, with both "strong" and "weak" forms. **(Section 2)**

2. We discuss the theoretical feasibility of these different conjectures. As we will see, strong test collapse is extremely unlikely, while weak test collapse is in principle possible but does not occur in practice. **(Section 2.1)**

3. We empirically confirm the finding of Papyan et al. (2020), that train-collapse occurs in many realistic settings. However, we find that test-collapse does not occur. **(Section 3)**

4. We show several settings where increasing train-collapse is anti-correlated with test performance, in both on-distribution and transfer-learning settings. This demonstrates that train-time neural collapse is not always desirable— and indeed, can be counterproductive— for some kinds of generalization. **(Section 4)**.

We thus conclude that Neural Collapse is primarily an *optimization* phenomenon, and its connections to generalization require further investigation.

## 1.1 Related Works

The Neural Collapse phenomenon was originally presented in Papyan et al. (2020), and led to a series of follow-up works investigating and extending it. Many of the subsequent works develop simplified models in which Neural Collapse on the train set can be theoretically proven and understood. For example, Fang et al. (2021) develops a "layer-peeled" model of training, and explores neural collapse in class-imbalanced settings. Mixon et al. (2020) proposes an alternate simplification, an "unconstrained features" model, in which train collapse also occurs. Wojtowytsch et al. (2020) and Zhu et al. (2021) also investigate the train collapse under unconstrained features model. Several works Poggio & Liao (2020a;b); Rangamani et al. (2021); Han et al. (2021) examine the Neural collapse with the square loss under different settings. Specifically, Poggio & Liao (2020a;b) give theory which predicts the properties of neural collapse for homogeneous, weight-normalized networks. Rangamani et al. (2021) proves that quasi-interpolating solutions obtained by gradient descent in the presence of weight decay have Neural collapse properties. Han et al. (2021) proposes

a generic decomposition of the MSE loss which, under certain assumptions, results in a simplified dynamical description (the "central path") which exhibits neural collapse on the train set. Lu & Steinerberger (2020) extend theoretical analysis of neural collapse to the cross-entropy loss (while previous works mainly considered MSE loss). They prove neural collapse on the train set in the "unconstrained features" setting. Ergen & Pilanci (2021) reformulate the last-layers of networks to convex formulations and give an explanation of Neural Collapse properties. Ji et al. (2021b;a) proposes unconstrained layer-peeled model which captures the properties of Neural Collapse and prove that gradient flow on this model converges to critical points exhibiting neural collapse in its global minimizer. Yang et al. (2022) and Pernici et al. (2021) observe that fixing the parameters (i.e. no back-propagation) in the final classifier as a simplex Equiangular Tight Frame (ETF) does not basically reduce the performance on the test set.

Liu et al. (2020) investigate the impact of a margin parameter added to softmax/cosine softmax loss in the setting of few-shot learning and they show that this margin parameter controls the degree of NC. They also show that a higher margin parameter (larger intra-class variance) leads to higher accuracy on the validation set, but lower accuracy on unseen classes at training time. Goldblum et al. (2020) study NC in the context of meta-learning, and the authors find that higher NC is better. They add a regularizer to the loss to increase NC appears to improve transfer learning results. Doersch et al. (2020) state that supervision collapse is an obstacle to learning good representations for few-shot learning. The definition of supervision collapse is the representations "represent only an image's (training-set) class, and discard information that might help with out-of-distribution classes", which is similar with the NC1 (feature collapse) definition. Roth et al. (2020) propose a strategy to reduce NC, and show that doing so improves the performance of deep metric learning. Kornblith et al. (2021) examine a variety of loss functions and find that loss functions that produce greater NC on the ImageNet training set sometimes get higher validation accuracy but transfer worse.

However, all the above papers present results for Neural Collapse on the train set, with an exception of Han et al. (2021), which gives a preliminary experiment on the test set collapse (see Figure 12 on page 20). They observe that "the rate of collapse is much slower on the test data compared to that on the train data", which agrees with our observations. Rangamani et al. (2021) has a discussion on neural collapse and generalization and argues that neural collapse is not related to good generalization as "Neural Collapse is a property of the dynamics independently of the size of the margin which provides an upper bound on the expected error". However, there has been no in-depth study on test data collapse and generalization.

One of the few papers focusing on collapse at test time is Galanti et al. (2021). Their work focuses on neural collapse for transfer learning, under a particular assumption: classes in the source and target tasks are selected randomly from the same class distributions. In contrast to their work, our transfer-learning experiment does not obey the assumptions on source/target task required by Galanti et al. (2021), since we consider a source task which is a "class-superset" of the target task. Another key difference is that Galanti et al. (2021) use a notion of "collapse" which only requires collapse to occur in the limit of infinite train size. However, we consider "collapse" to occur if it occurs at finite train size. This finite-sample definition follows the original framework of Papyan et al. (2020), and is essential to a meaningful definition of collapse. We elaborate on this important point in Section 2. At first glance, the conclusion in the transfer learning setting arrived at by Galanti et al. (2021) contradicts to ours results. One reason for this can be the class number in their pretraining is larger than the downstream tasks, while we are considering a "super-class" setting where the downstream task has finer labels and larger class number. We pre-train on large datasets while fine-tune on tasks with limited data. This is a standard transfer learning setting where transfer learning is particularly useful in practice.

Zhu et al. (2021) and Mixon et al. (2022) provide empirical evidence that neural collapse can happen for training data with random labels. However, the presence of neural collapse on training data cannot indicate whether the network generalizes or not. Feng et al. (2021) points out that negligence of valuable intra-class semantic difference is the reason for worse transferability of existing supervised pre-training methods, compared with the powerful transferability of self-supervised pre-training. They propose a new supervised pre-training method based on Leave-One-Out K-Nearest-Neighbor to preserve part of intra-class difference, i.e. to have less neural collapse. Extensive empirical studies show their method leads to better transferring to downstream tasks. Their conclusion agrees with ours. Inspired by the property of nearest-class center decision rule, Galanti (2022) proposes "minimal NCC-depth" to capture the relationship of neural collapse

and generalization, as they also observe no clear relation between training data collapse and generalization. Note that the class-distance normalized variance (CNDV) definition used in this paper comes from Galanti (2022) is from Galanti et al. (2021) and it is essentially the same as the definition in Papyan et al. (2020). In contrast, our NC1 definitions explicitly consider the dependency on train set size, which is more precise than the CNDV definition and the original definition of Papyan et al. (2020). Also, we extend the NC1 definition to the test set, with both "strong" and "weak" forms. For the empirical findings, on the difference between train set and test set collapse, the observations in Figure 1 of Galanti (2022) is similar to ours, i.e. test-collapse seems to occur to a much less extent than train-collapse. Our main contribution is making this observation mathematically precise, by considering the asymptotic limit as a function of train samples (which has not been done in prior work as far as we are aware).

## 1.2 Notation

Let $\mathcal{X}$ be the input space, and $\mathcal{Y}$ be the label space. We consider multi-class classification problems, where $\mathcal{Y} = [k]$ for some $k \in \mathbb{N}$. Let $\mathcal{D}$ be the target distribution over $\mathcal{X} \times \mathcal{Y}$. *Training procedures*[1] are functions which map a train set $S \in (\mathcal{X} \times \mathcal{Y})^n$ and an iteration count $t \in \mathbb{N}$ and to a model $f$. In this work, we will always consider Stochastic-Gradient-Descent (SGD)-based training procedures, where $t$ is the number of SGD steps. For a fixed train set $S$ of size $n$, let $f_S^t$ denote the model output by the training procedure after $t$ iterations. So $\mathsf{Train} : (S, t) \mapsto f_S^t$, where $\mathsf{Train}$ denotes the training procedure. For a given model $f_S^t$, let the *last-hidden-layer feature map* be denoted $h_S^t : \mathcal{X} \to \mathbb{R}^d$. This is the feature-map induced by the trained model, as a map from inputs into $\mathbb{R}^d$.

## 2 Defining Neural Collapse

We first define two kinds of Neural Collapse: on the train set, and on the test set. Our definitions naturally extend the definitions in Papyan et al. (2020), but are more precise since we explicitly include the train/test distinction, and the dependency on training iterations $t$ and train samples $n$. This is essential to describe the relevant asymptotic limits in the "collapse".

Throughout this work, we focus only on the first conjecture from Papyan et al. (2020): "NC1 (Variability Collapse)." NC1 captures the within-class variance and it is the most relevant one to generalization. Also, the subsequent Simplex ETF conjecture is particularly meaningful only if NC1 is true, that is features cannot collapse to a simplex ETF if the variability does not "collapse" at all. When we refer to "neural collapse" in this work, we specifically are referring to "variability collapse." We first define collapse on the train set, which follows closely the definition in Papyan et al. (2020).

**Definition 1** (Train-Collapse)**.** *For a particular train set $S$, we say a training procedure $T$ exhibits Train-Collapse on $S$ if there exists some distinct $\mu_1, \mu_2, \ldots, \mu_k \in \mathbb{R}^d$ such that*

$$\forall (x_i, y_i) \in S : \quad \lim_{t \to \infty} h_S^t(x_i) = \mu_{y_i}$$

That is, the trained network converges to representations such that all train points of class $k$ get embedded to a single point $\mu_k$ (called the "class means" in Papyan et al. (2020)). The conjecture below then states conditions under which Train-Collapse occurs. This conjecture is meant to capture the original NC1 conjecture of Papyan et al. (2020), which was demonstrated empirically across many settings.

**Conjecture 2** (Train-Collapse Conjecture, informal)**.** *For all train sets $S$ containing at least two distinct labels, and all training procedures $T$ corresponding to SGD on "natural" sufficiently-deep and sufficiently-large neural network architectures: $T$ exhibits Train-Collapse on $S$.*

Crucially, we state Conjecture 2 for train sets of *all sizes*. This dependency on train set size is implicit, but omitted from Papyan et al. (2020) — it will become especially important when we discuss generalization, and this makes the biggest difference from the CDNV definition given by Galanti et al. (2021), which assume infinite train size. This behavior is called a "collapse" because regardless of the train set size, any big-enough

---

[1]We can consider randomized training procedures by allowing an additional random string as input. We omit this randomness throughout, for notational clarity.

network that enables neural collapse converge to this discrete limiting structure. We replicated this finding in most of our experiments. However, for completeness we acknowledge that this conjecture does not hold fully universally, and there are subtleties in practice[2]. Nevertheless, we believe the NC1 conjecture captures the right qualitative behavior in many realistic settings.

We also acknowledge that Conjecture 2, while more precise than the conjectures in Papyan et al. (2020), is still not fully formal. For example, it only applies to "natural" architectures and not all architectures, and does not quantify what "sufficiently large" means. In our experiments, we also apply weight decay, batch normalization (BN), tune different learning rates for each model. Rangamani & Banburski-Fahey (2022) shows that neural collapse does not necessarily happen when training without weight decay and without biases. Ergen & Pilanci (2021) study the connection between NC and BN. Also, Galanti (2022) shows that depth also matters for NC to happen. This restriction to "natural" architectures is a known obstacle to formalism in deep learning theory (e.g. Nakkiran (2021)) and is necessary to avoid pathologies such as Abbe & Sandon (2020). Nevertheless, our definitions take a step towards greater formalism, and this precision will be useful in understanding connections to generalization. Refining our definitions and conjectures further is an area for future work.

The notion of train-collapse described above (and in Papyan et al. (2020)) is an *optimization* notion: it involves only behavior of a model on its train set, and not behavior at test time. Thus, it is a priori unclear whether this notion is related to generalization aspects of models. To explore this, we first extend the definition of Neural Collapse to the test set, and then investigate whether this test-collapse occurs in practice. The most immediate way to formulate test collapse is to use the exact same formulation and quantifier on sample size $n$ with Train-Collapse. We call this similar formulation with Train-Collapse Strong Test-collapse.

**Definition 3** (Strong Test-Collapse). *A training procedure $T$ exhibits Strong Test-Collapse on distribution $\mathcal{D}$ if for all sample sizes $n \in \mathbb{N}$, the following holds with probability $1$ over sampling $S \sim \mathcal{D}^n$: there exists some distinct $\mu_1, \mu_2, \ldots, \mu_k \in \mathbb{R}^d$ such that*

$$\text{with prob } 1 \text{ over } (x, y) \sim \mathcal{D}: \quad \lim_{t \to \infty} h_S^t(x) = \mu_{y^*(x)}$$

*where $y^*(x) := \arg\max_y p_{\mathcal{D}}(y|x)$ is the Bayes-optimal classification under distribution $\mathcal{D}$.*

Strong Test-Collapse requires that test points $x$ map to their "correct" embedding point $\mu_i$, where $i$ is the Bayes-optimal class for $x$. However, unless $n$ is large enough that we are able to learn the Bayes-optimal classifier exactly, Strong Test-Collapse will not occur. Since this natural extension from Train-Collapse is hard to happen, we define a "weak" version of test set collapse which is likely to happen. It requires only that test points embed as *one of $k$* discrete points $\mu_1, \mu_2, \ldots \mu_k$, without requiring that all points of class $i$ map to $\mu_i$.

**Definition 4** (Weak Test-Collapse). *A training procedure $T$ exhibits Weak Test-Collapse on distribution $\mathcal{D}$ if for all sample sizes $n \in \mathbb{N}$, the following holds with probability $1$ over sampling $S \sim \mathcal{D}^n$: there exists some distinct $\mu_1, \mu_2, \ldots, \mu_k \in \mathbb{R}^d$ such that*

$$\text{with prob } 1 \text{ over } (x, y) \sim \mathcal{D}: \quad \lim_{t \to \infty} h_S^t(x) \in \{\mu_i\}_{i \in [k]}$$

There are several important differences between the notions of test-collapse and train-collapse. First, for test-collapse we require that the train set $S$ is not arbitrary, but sampled from some distribution $\mathcal{D}$. And we check for limiting behavior with respect to *new* samples from $\mathcal{D}$, as opposed to train samples from $S$. However, both train and test collapse require the collapse to occur *for all finite sample sizes $n$*, letting only time $t \to \infty$. This is the meaningful asymptotic, since taking limit of samples $n \to \infty$ would obscure almost all aspects of learning, which is most interesting at finite-sample sizes.

With the dependency on train set size, which is crucial when discussing generalization, our definitions are a natural extension of definitions given in Papyan et al. (2020) and Galanti et al. (2021), and they are a step forward to evaluate the correlation of neural collapse and generalization.

---

[2]For example, we found in some settings training variability does not collapse to negligible value, such as CIFAR-10 and STL-10 dataset with VGG architectures (see Figure 3) . In some preliminary experiments we also found that adding stochasticity (such as dropout noise) often accelerated collapse, which is consistent with the theoretical model in Papyan et al. (2020).

### 2.1 Remarks on Feasibility

With the above definitions, we can see that strong test-collapse is too strong a property to apply in realistic settings. We discuss this infeasibility here, and then corroborate this with experiments in the following section.

**Infeasibility of Strong Test-Collapse.** First, note that both train-collapse and test-collapse definitions require that collapse occurs for all train set sizes $n \in \mathbb{N}$. This property is easy to satisfy for train-collapse, but is an extremely strong property for test-collapse. In particular, the "strong" form of test collapse (Definition 3) is too strong to hold in practice: it implies that a Bayes-optimal classifier can be extracted from the trained model features, even if the model is trained on only e.g. $n = 10$ samples. Even with large but finite $n$, it's hard to learn Bayes-optimal classifier exactly (and it is unlikely to happen in most realistic settings). This is because, according to Definition 3, the representation must map test inputs to their "correct" cluster, and thus the correct label can be extracted from the cluster identity.

However, the "weak" form of NC1-test (Definition 4) still has hope of holding, since it does not imply learning a Bayes-optimal classifier. Nevertheless, note that even the "weak" form is a fairly strong condition for neural networks: it implies that trained networks (on *any* size train set) learn feature-maps $h$ such that the push-forward $h_*(\mathcal{D})$ is a discrete measure. Mapping the continuous measure $\mathcal{D}$ to a discrete measure is a strong property, and one that is unlikely to hold for standard neural networks.

**Feasibility of Weak-Collapse.** While weak-collapse is unlikely to hold for neural networks trained with SGD, the definition itself is non-vacuous: there exist learning methods which are "reasonable" (asymptotically consistent) and exhibit weak test-collapse. To see this, consider the following modified training procedure: first, train a neural network as usual to get a network $f : \mathcal{X} \to \mathcal{Y}$. Then, construct another network $f'$ such that the *last-layer representation* of $f'$ is a one-hot encoding of the *classification decision* of $f$. That is, the representation $h'(x) \in \mathbb{R}^k$ satisfies $h'(x) := \vec{e}_{f(x)}$ where $\{\vec{e}_i\}$ are standard basis vectors. This can be constructed by, for example, adding post-processing layers to $f$. Now, the training procedure which outputs $f'$ will satisfy weak test-collapse of its representations, since its representations are always one of the $k$ standard basis vectors by construction.

**Desirability of Neural Collapse for Generalization.** Armed with these definitions, we can now consider whether train or test collapse are necessary or sufficient for on-distribution generalization. First, neither train nor test collapse are strictly necessary for good generalization: As discussed, it is possible to construct models with identically good generalization performance, but which satisfy neither train nor test collapse. There are even natural, non-contrived examples of this: models trained for less than one epoch (the "Ideal World" in the terminology of Nakkiran et al. (2020)) will not exhibit train collapse, because they are not trained to fit their train set. And yet, as demonstrated in Nakkiran et al. (2020), they can match the performance of interpolating models. This "one epoch" regime is also relevant in practice, where models are trained on massive data sources such as internet scrapes, often for less than one epoch (Brown et al., 2020; Raffel et al., 2020; Komatsuzaki, 2019).

Further, neither train collapse (Definition 1) nor weak test-collapse (Definition 4) are sufficient for generalization. It is possible to construct models which satisfy train collapse perfectly, but which are random functions at test time. Likewise, it is possible to construct models which satisfy weak test-collapse, but have random classification decisions.

Strong test-collapse (Definition 3) *is* sufficient for good test performance, since it implies that test inputs map to the "correct" cluster in representation-space. However, as we discussed, strong test-collapse is infeasible, and impossible in practice.

## 3 Experiments: Train and Test Collapse

Here we complement our theoretical discussion by measuring both train and test collapse in realistic settings, following the experiments of Papyan et al. (2020). We find that train-collapse occurs in many settings, while

test-collapse (both strong and weak) does not. We also show the dependency on the train set size: larger train sets lead to *stronger* test collapse, but *weaker* train collapse. Note that we say stronger collapse or more collapse when the feature variance is smaller. This further highlights the importance of distinguishing between the two forms of collapse, since they can be anti-correlated. Also, stronger train set collapse can lead to worse test performance, which means that stronger collapse on train set itself is not correlated with better generalization.

## 3.1 Measuring Collapse

It is not possible to measure collapse strictly according to Definitions 1, 3 and 4, since they involve a $t \to \infty$ limit. Instead, we follow exactly the experimental procedure of Papyan et al. (2020), and measure approximations which capture the "degree of collapse." We restate their procedure here for convenience. Measuring collapse require finding the vectors $\mu_1, \mu_2, \ldots \mu_k \in \mathbb{R}^d$, which embeddings collapse to. The choice of these vectors depends on the setting, as below.

**Train Collapse.** For the train set, $\mu_i$ is defined as the train class-means:

$$\hat{\mu}_i := \mathbb{E}_{(x,y) \in S}[h_S^T(x) \mid y = i]$$

where $T$ is the maximum train time in the experiment. Define the global mean as $\hat{\mu} := \sum_i \hat{\mu}_i / |\mathcal{Y}|$. Then, the "degree of train collapse" is measured as:

$$\mathsf{TrainVariance}(t) := \frac{\mathbb{E}_{(x,y) \in S}[||h_S^t(x) - \hat{\mu}_y||^2]}{\mathbb{E}_i[||\hat{\mu}_i - \hat{\mu}||^2]}$$

Smaller values of this quantity indicate more "collapse." The numerator here is the "within-class variance" and it is normalized by the "between-class variance", in the terminology of Papyan et al. (2020). This definition follows the experimental measurements in Papyan et al. (2020).

**Strong Test Collapse.** For test collapse, $\mu_i$ is defined as the test class-means:

$$\bar{\mu}_i := \mathbb{E}_{(x,y) \sim \mathcal{D}}[h_S^T(x) \mid y = i]$$

The global mean is $\bar{\mu} := \sum_i \bar{\mu}_i / |\mathcal{Y}|$. Then, the "degree of strong test collapse" is measured as:

$$\mathsf{StrongTestVariance}(t) := \frac{\mathbb{E}_{(x,y) \sim \mathcal{D}}[||h_S^t(x) - \bar{\mu}_y||^2]}{\mathbb{E}_i[||\bar{\mu}_i - \bar{\mu}||^2]}$$

**Weak Test Collapse.** For weak test-collapse (Definition 4), we do not require that representations collapse to their *class means*, but simply to some $\mu_i$. Thus, we define $\{\widetilde{\mu}_i\}$ as the result of $k$-means clustering on the following set of vectors: $\{h_S^T(x)\}_{x \in \text{TestSet}}$. The global mean is $\widetilde{\mu} := \sum_i \widetilde{\mu}_i / |\mathcal{Y}|$. And the "degree of weak test collapse" is measured as:

$$\mathsf{WeakTestVariance}(t) := \frac{\mathbb{E}_{(x,y) \sim \mathcal{D}}[\arg\min_{i \in [k]} ||h_S^t(x) - \widetilde{\mu}_i||^2]}{\mathbb{E}_i[||\widetilde{\mu}_i - \widetilde{\mu}||^2]}$$

## 3.2 Experimental Results

**Setup.** We consider image classification tasks with MNIST, FashionMNIST, CIFAR-10, SVHN and STL-10 datasets. We train Resnet, DenseNet and VGG networks with stochastic gradient descent (SGD) to minimize the cross-entropy loss. All tasks were trained on a single GPU with batch size 128 and 80000 SGD iterations. See Appendix A for more details and references about the datasets, architectures and training mechanisms.

In the following of this section we show that the test collapse does not occur with experiments on a wide range of datasets and model architecture combinations. We show that train collapse and test collapse can be anti-correlated and more train collapse can lead to worse test performance. Considering the dependency on train set size is fundamental to generalization.

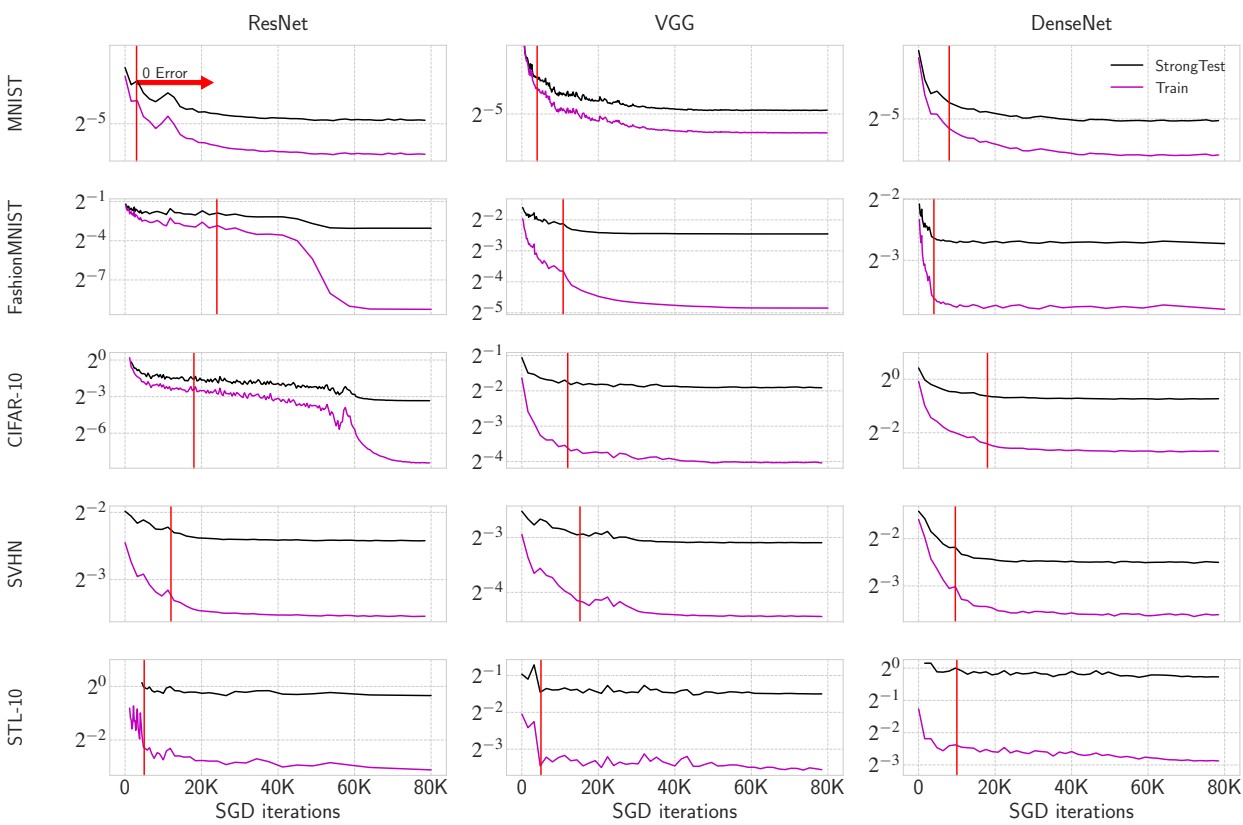

Figure 3: **Failure of Test Collapse.** Training and test variance vs. SGD iterations, for various dataset and architecture combinations. All test sets (black line) do not collapse to negligible variance, and have much less collapse than the train sets (purple line).

**Failure of Test Collapse.** In Figure 2, we train a single model (ResNet-18 on CIFAR-10) and measure TrainVariance, WeakTestVariance, and StrongTestVariance as a function of train time $t$. That is, we measure the degree of train and test collapse over increasing time. We see that train collapse appears to occur, while test variance does not decrease to negligible value. In particular, there is a "generalization gap" in the Train vs. Test Variances: the TrainVariance appears to converge to 0 as $t \to \infty$, while TestVariance (both weak and strong) do not. For the remainder of the experimental results, we plot only "strong" test collapse, since we generally observe that both strong and weak collapse have similar behavior.

In Figure 3, we train different models on various datasets and measure TrainVariance and StrongTestVariance as a function of train time $t$. We train all models to get 0 training error and continue training to achieve close to 0 training loss[3]. We see Strong Test-Collapse does not occur on all settings, and has a large gap with Train Collapse. Again, the results show that Neural Collapse is mainly an optimization phenomenon and not a generalization one: test set does not collapse to negligible value in any setting, together with our theoretical argument of infeasibility of Strong Test-Collapse in 2.1, we claim a failure of test collapse.

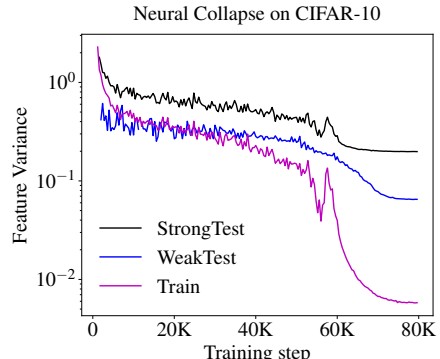

Figure 2: **Neural Collapse on CIFAR-10.** Collapse occurs on the train set, but not on the test set (neither Strong nor Weak).

---

[3]We use "close to 0" to mean when the loss is below $10^{-5}$.

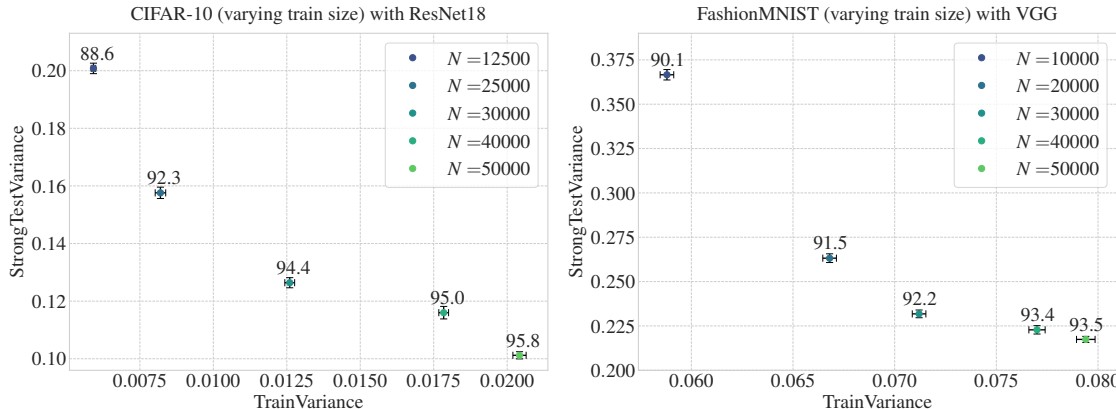

Figure 4: **Train vs. Test Anti-Correlation.** The numbers up the dots are corresponding test accuracy of different train set size ($N$). We observe that train and test collapse are anti-correlated and small train variance has worse test accuracy. That is neural collapse can hurt generalization. *Top:* ResNet18 trained on subsets of CIFAR-10. *Bottom:* VGG11 trained on subsets of FashionMNIST.

**Train vs. Test Anti-Correlation.**  In Figure 4 we train a ResNet18 on CIFAR-10, and vary the size of the train set from $N = 12500, 25000, 30000, 40000$ to $N = 50000$. We also report results on training a VGG11 network with batch normalization on different subsets of FashionMNIST. For each train set size $N$, we have 5 runs with different random seeds (the subset of each run is different because of random selection), and report the average of the variance and plot the error bar.

We train all models past the point of 0 training loss and stop training when the training loss decreases to $10^{-6}$ in each run. Figure 4 plots the train collapse (TrainVariance) compared to the test collapse (StrongTestVaraince) at the end of training, for different train set sizes. We also report the corresponding test accuracy right up the dots in Figure 4. We find that as the train set size increases, the train variation increases (less train collapse), however, the test accuracy gets higher and the test variation decreases (more test collapse). This illustrates that test and train collapse are not always correlated, and thus it is important to distinguish between the two: "better" optimization behavior (train set collapse) accompanies worse generalization behavior. Also, by considering the dependency on train set size, we see that stronger train set collapse itself does not imply good generalization. If you look at the train variance in x-axis and the accuracies up in the dots in Figure 4, it shown that solutions that does not exhibit train collapse (larger train variance) actually have good generalization (better test accuracy). Also, this observation matches the claim in Nakkiran et al. (2020) which say "1-epoch" CIFAR models have similar performance as multi-epoch models but the "1-epoch" models do not exhibit train collapse.

One limitation of this experiment is that we evaluate collapse at finite train time, and not at $t = \infty$. Indeed, at $t = \infty$ we expect the train variation to be identically 0 for all data sizes (by the definition of collapse), but the test variation to decay with larger data sizes. This situation is analogous to measuring train/test error itself for overparameterized models: for large enough models, train error will always be 0, but test error will decay with the data size. This experiment thus highlights the importance of measuring both train & test quantities, and the subtlety involved in measuring collapse at finite time.

We also acknowledge that in this experiment, increasing the size of the train set is correlated with both better test collapse, and better generalization. However, we caution that this should not be seen as evidence that test collapse is mechanistically related to generalization. First, because the test variance does not truly "collapse", it just reduces, as already discussed. And second, because this reduction in test variance is in some sense necessary for any model with improved test error— since high test variance would produce noisy classification decisions. Thus, the correlation of test variance and generalization in this experiment should not be surprising. We are cautious to make the claim on the correlation of test set collapse and generalization, and we think this needs more careful study. We think it's better to leave this question in a separate work, as

this work mainly focuses on the correlation of train set collapse and generalization and most previous works on neural collapse focus on the train set.

## 4 Collapsed Features Transfer Worse

In the previous section, we showed that train-time collapse can be anti-correlated with generalization performance, when measuring generalization on-distribution. We now explore generalization on other distributions, by considering transfer learning. We consider the standard transfer-learning setting, where models are usually pre-trained on massive datasets and then fine-tuned for tasks on small datasets. Now we investigate generalization on downstream tasks, to understand the role of Neural Collapse in *transfer-learning* and *representation learning.*

### 4.1 Test Collapse implies Bad Representations

We first observe that, using our definition of test collapse, a model which has fully test-collapsed will have representations that are bad for most downstream tasks. To see this, consider the following example. Suppose we have a distribution $\mathcal{D}$ with ten types of images (as in CIFAR-10), but we group them into two superclasses, such as "animals" and "objects." We then train a classifier on this binary problem (e.g. CIFAR-10 images with these binary labels). Let the feature map of the fully-trained model (that is, the limiting model as $t \to \infty$) be denoted $h$. If this model exhibits even weak test collapse, then there exist vectors $\{\mu_1, \mu_2\}$ such that the representations satisfy:

$$\Pr_{x \sim \mathcal{D}} [h(x) \in \{\mu_1, \mu_2\}] = 1. \tag{1}$$

That is, the representations will by definition "collapse": every input $x \sim \mathcal{D}$ will map to exactly one of two points $\mu_1, \mu_2$. This property is clearly undesirable for representation learning. For example, suppose we use these representations for learning on a related task: the original 10-way classification problem. It is clear that no classifier using the fixed representations (linear probing scheme) from $h$ can achieve more than 20% test accuracy on the original 10-way task: each group of 5 classes will collapse to a single point after passing through $h$ (by Equation (1)), and will become impossible to disambiguate among these 5 classes. This example is formalized in the lemma below.

**Lemma 1.** *Let $(x, y) \sim \mathcal{D}$ be any target distribution defining a balanced $2k$-wise classification task. Let $\mathcal{D}_2$ be the pretraining distribution, defined by super-classing $\mathcal{D}$ into a balanced binary classification task. That is, the distribution of $\mathcal{D}_2$ is given by $(x, F(y))$ for $(x, y) \sim \mathcal{D}$ and some balanced partition $F : [2k] \to \{0, 1\}$.*

*Now let $h : \mathcal{X} \to \mathbb{R}^d$ be pretrained representations that are fully test-collapsed with respect to $\mathcal{D}_2$, with WeakTestVariance exactly 0.*

*Then, all classifiers that input only representations $h(x)$ have test accuracy at most $1/k$ on $\mathcal{D}$. Formally, for all functions $f : \mathbb{R}^d \to \{0, 1, .., 2k - 1\}$, the test accuracy*

$$\Pr_{(x,y) \sim \mathcal{D}} [f(h(x)) = y] \leq \frac{1}{k}.$$

As Weak Test-collapse is a weaker condition than Strong Test-collapse, Lemma 1 also holds for Strong Test-Collapse. The proof of Lemma 1 is straightfoward, and included for completeness in Appendix C. In our experimental results below, we show the variance of Strong Test-collapse. Even if we do not fix the pre-trained representations and fine-tune all the parameters (fine-tuning scheme), as shown in our experimental results 4.2, fully collapsed features (less within-class variance) lead to worse down-stream task performance. The results in Feng et al. (2021) show similar observations with ours and their proposed method to enlarge within-class variance actually improves test performance. This shows that test collapse is undesirable for even an extremely simple transfer learning task (where we transfer to the same distribution, with finer label structure). In the following sections, we will demonstrate this result experimentally, even for classifiers which have not fully collapsed.

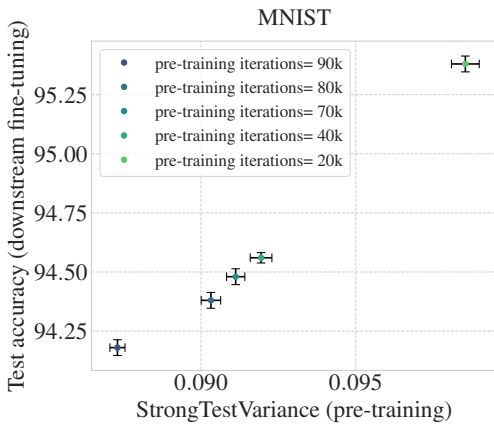 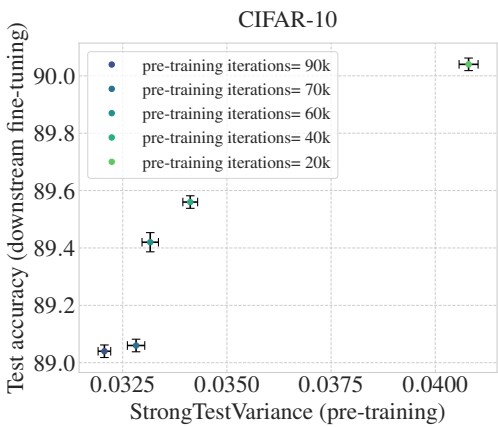

Figure 5: **Collapsed Features Transfer Worse.** We save different checkpoints during pre-training, and use them to initialize the downstream models. We fine-tune all the parameters of the model. The $x$-axis shows the StrongTestVariance of those checkpoints on the pre-training test set, and $y$-axis shows the test accuracy after fine-tuning on downstream tasks. We find that stronger test collapse (i.e. lower variance) is correlated with lower downstream test accuracy. *Left:* MNIST with a 3 hidden layer fully-connected network. *Right:* CIFAR-10 with a standard Resnet18.

## 4.2 Experiments

There are many relevant settings in transfer learning, especially in practice. For example, in practice, we often pre-train on a "generic" task with massive datasets, and then fine-tune on a specific task with limited data. This specific task may involve finer-grained labels than the generic task, which parallels our experimental setup. We train a 3 hidden layer fully-connected networks with 1024 units per layer on MNIST, and a standard Resnet18 on CIFAR-10. For pre-training, we use a subset of the train set and perform 2-class classification (via super-classing). For *fine-tuning*, we use the weights pre-trained as initialization of the weights other than the last classification layer, and do standard (10-class for MNIST, and 8-class for CIFAR-10) classification with a much smaller held-out subset. We do not report results with linear probing, as it gives much worse transfer-performance than *fine-tuning* scheme. See more details in Appendix B.

Here we show transfer learning results on MNIST and CIFAR-10. To see the correlation between neural collapse (on test set) and generalization, we plot the degree of test set collapse during pre-training and test performance in down-stream tasks. We report the average of 5 runs with different random seeds, and give the error bars, as illustrated in Figure 5. We see that for both MNIST and CIFAR-10, the checkpoints with more Test Collapse gives worse transfer-performance on downstream tasks. That is, in these settings more Test Collapse actually leads to learning worse features. This demonstrates that neural collapse does not always lead to good representation learning— when the class number in pre-training is less than the number of downstream tasks. collapse actually harms representation quality.

## 5 Conclusion

We show that Neural Collapse is primarily an optimization phenomenon, and does not always correlate with better generalization. We propose more precise definitions— "strong" and "weak" Neural Collapse for both the train set and the test set— which disentangle generalization and optimization behaviors. We believe these more precise definitions aid in clarifying the literature around neural collapse, and will help guide further study. By investigating the train and test collapse on various dataset and architectures, we show that while train collapse reliably occurs in many settings, test collapse does not. Our theoretical formulations and empirical observations suggest that while neural collapse continues to be an intriguing phenomenon and a promising optimization research program, its relevance to generalization requires further study.

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

## A    Experimental setup for Figure 3

**Datasets.** We consider image classification tasks with MNIST LeCun et al. (1998), FashionMNIST Xiao et al. (2017), CIFAR-10 Krizhevsky et al. (2009), SVHN Netzer et al. (2011) and STL-10 datasets Coates et al. (2011). SVHN was sub-sampled to $N = 4600$ samples per class as training set and $N = 1500$ samples per class for test set. Other datasets are following the standard setup. No data argumentation was done and we pre-process the images pixel-wise by subtracting the mean and dividing by the standard deviation.

**Models.** We train standard Resnet18 and DenseNet201 for MNIST, FashionMNIST, CIFAR10 and SVHN. Resnet50 and DenseNet201 were trained for STL10. For all datasets we also train VGG11 with batch normalization. All models were trained from scratch with open source code from torchvision models.

**Optimization mechenism.** We use stochastic gradient descent (SGD) with momentum 0.9 and minimize the cross-entropy loss. All tasks were trained on a single GPU with batch size 128 and 80000 SGD iterations. Initial learning rate is 0.1 for Resnet18 and Resnet50 and 0.01 for DenseNet201 and VGG architectures. We decay the learning rate with cosine annealing scheme.

## B    Experimental setup for transfer learning

**Super-class pre-training.** For MNIST, we set all odd numbers as one class and all even numbers as the other class. We train the model with the first $N = 1000$ training samples as train set and the first $N = 200$ test samples as test set. For CIFAR-10, we combine samples of 'airplane, automobile, ship, truck' as one (objects) class and 'bird, cat, frog, horse' as the other (animals) class. The two classes are balanced and have 40000 training samples, and 8000 test samples. We use a subset with $N = 20000$ training samples (to keep each class balanced, we randomly choose 2500 samples from each original class) and $N = 4000$ (500 samples from each original class) test samples for pre-training. The learning rate for MNIST with fully-connected networks is 0.001 while for CIFAR-10 with ResNet18 is 0.1. We decay learning rate with cosine annealing scheme. The models were trained minimizing the cross-entropy loss using SGD with momentum 0.9 for 100000 SGD iterations.

**Fine-tuning.** We initialize the weights (other than the last classification layer) of the downstream task with the pre-trained weights and fine-tune the whole network. For MNIST, we do the standard 10-class classification, while we sample another 500 samples from training set for training and 100 samples from the test set for inference. For CIFAR-10 we implement a 8-class classification ('airplane, automobile, ship, truck, bird, cat, frog, horse') with another 10000 training samples as train set and another 2000 test samples as test set. The optimization methodology is the same as in pre-training, other than the learning rate. We search over 0.0005 to 0.25 in fine-tuning for both MNIST and CIFAR-10 and report the best test accuracy of all swept learning rates.

## C    Proof of Lemma 1

*Proof.* As given in Lemma 1, for $(x_1, 0) \sim \mathcal{D}_2$ and $(x_2, 1) \sim \mathcal{D}_2$, there have $h(x_1) = \mu_1$ and $h(x_2) = \mu_2$, where $x_1$ and $x_2$ are samples from class 0 and class 1 correspondingly. For a test sample $x \sim \mathcal{D}$, which is sampled from the target distribution, the corresponding feature representation $h(x)$ also maps $x$ to $\mu_1$ or $\mu_2$, due to the superclass property.

For any function $f : \mathbb{R}^d \to \{0, 1, ..., 2k - 1\}$, which maps the feature representation $h(x) \in \mathbb{R}^d$ of a sample $x$ to class index $\hat{y}$,

$$\hat{y} = f(h(x)) = \begin{cases} f(\mu_1) = k_1 & \text{if } x \text{ is superclassed to class 0} \\ f(\mu_2) = k_2 & \text{if } x \text{ is superclassed to class 1} \end{cases}$$

where $k_1, k_2$ are class index from $\{0,1,.., 2k\text{-}1\}$. As we are considering the class-balanced case, for each sample $x \sim \mathcal{D}$, the probability of its true label $y$ to be $k_1$ or $k_2$ is $Pr(y = k_1) = \frac{1}{2k}$ and $Pr(y = k_2) = \frac{1}{2k}$. Then

$$\Pr_{(x,y)\sim\mathcal{D}}[\hat{y} = y] = \begin{cases} \dfrac{1}{k} & \text{if } y = k_1 \text{ or } y = k_2 \\ 0 & \text{if } y \neq k_1 \text{ and } y \neq k_2 \end{cases}$$

Hence

$$\Pr_{(x,y)\sim\mathcal{D}}[f(h(x)) = y] \leq \frac{1}{k}$$

and the equality holds when $y = k_1$ or $y = k_2$. $\qquad\square$

