# OpenReview forum: "Limitations of Neural Collapse for Understanding Generalization in Deep Learning"
_TMLR — Rejected by TMLR_

### Review · Reviewer_Lq57 · 2023-01-30

**Summary Of Contributions:**

This paper studies the problem of Neural Collapse (NC), a phenomenom in which a neural network learns to map the input data to a finite set of outputs. In this work, the authors extend the definition originally provided by Papyan et. al. (2020) to include the dataset size, and define two variations of said definition to work on test data (instead of training data).

Based on these new definitions, the authors conduct a number of experiments to understand the correlation between training and test neural collapse, as well as the relation between representation learning and test neural collapse. The experimental results show that: i) there can exist a negative correlation between training and test neural collapse; ii) that test neural collapse is less likely to ocurr; and iii) that having neural collapse is not always desirable to obtain better representations.

**Audience:**

Yes

**Broader Impact Concerns:**

None.

**Claims And Evidence:**

No

**Requested Changes:**

**Critical to secure my recommendation**
1. Overfitting should be ruled out of the equation or, at least, the relantionship between overfitting and neural collapse should be clarified.
2. The definition of weak test collapse should be removed or properly used in the paper.
3. Claims need to be tone down.
4. Further effort should be put on the "not collapsing" behaviour on the test dataset in figure 3, as all trends are identical.
5. Use proper and strong arguments for the decisions taken in the paper, specially those regarding arbitrary decisions taken on the proposed definitions.
6. Properly investigate why results from existing work contradict the results from the manuscript.

**It would strengthen the paper**
1. Figure 1 is unecessary and it is cited in page 4.
2. Remove unnecesary maths and clarify the rest.

**Strengths And Weaknesses:**

**Strenghts:**
1. The motivation of the paper is clear: NC was only defined in the training dataset, and there is no intuition of what this quantity means when applied to the test dataset (yet prior works have implied generalization properties to NC).
2. The paper is clearly written and easy to follow.
3. The work is some interest to the community.

**Weaknesses:**
1. Mathematical definitions are overly complicated (e.g., definition 3 is the definition of almost-sure convergence), and there is no need to have the statement in lemma 1 as a lemma.
2. Some claims are overstated and/or unjustified. For example "This property is clearly undesirable for representation learning" after Eq. 1. Good discrete representations in a neural network is something I would personally love to achieve.
2. The definition of weak test-collapse is quite arbitrary, and not used in the paper.
3. Some claims are rather weak and artificial. For example;
   1. The argumentation on the feasibility of strong-test collapse in section 2.1: it argues against the feasibility of learning a Bayes optimal classifier but: i) that classifier was arbitrarily introduced in Def. 3 (for the experimental part, the labels could've been used), and ii) the more-suitable weak-test collapse then needs to rely on k-means (which falls into local optima) to measure its performance.
    2. The example to start the argument of section 4.1 ("Test Collapse implies Bad Representation") is arbitrary and misleading. To defend the opposite argument, one could build a similar example where the model is pre-trained on a finer set of labels that the ones tested on.
3. Important questions (such as why their results contradict the ones from the related work section, especially Galanti et al. (2021)), are conjectured/glossed-over rather than systematically studied.
4. I have serious concerns regarding the experimental setup, and thus the claims within the paper. Specifically:
    1. Every result in the manuscript yells overfitting to me (either in its usual form for section 3.3, or in its meta-learning form for section 4.1). More TrainVariance seems to imply better generalization and more StrongTestVariance (pre-training) implies better ood generalization because it is overfitting less.
    2. The StrongTest and Train lines in every plot from figure 2 and 3 are really suspicious, as the exact same trends (increasing, decreasing, bumps, etc) appear in both exactly at the same iterations. To me it looks like there is a constant term such that $\log_2 \text{Train} \propto \log_2 \text{StrongTest}$.
    3. From the experimental section, it seems pretty clear that train variance is to generalization what test variance is to ood generalization. In this regard, I find figures 4 and 5 rather misleading again, as plotting the test accuracy in the y-axis in figure 4 would show the same trend in both figures (or, equivalently, showing in figure 5 in-distribution test variance and out-distribution test variance).

---

### Review · Reviewer_1Pao · 2023-02-01

**Summary Of Contributions:**

The NC phenomena (Papyan, Han, and Donoho (2020) ) have received considered amount of attention. However, some claims from the original paper are not precise and need further investment, in particular, the relation between NC and generalization/representation learning.
The main goal of the paper is to clarify, through more precise definitions and some experimental results, that NC is mainly an optimization concept and does not necessarily lead to better generalization/features. I mostly agree with this message, and I also think many researchers in this area are well aware of the limitation of NC (e.g., see https://twitter.com/jasondeanlee/status/1494532192772542466). The current paper, in some sense, helps boost this signal.

The claimed contribution of the paper includes:
1. More precise definitions of NC for both train/test.
2. Clarify that NC can happen for the training set but usually not for the test set.
3. Some empirical evidence that there isn't direct relation between NC and generalization/better representations.


**Audience:**

Yes

**Claims And Evidence:**

No

**Requested Changes:**

In contribution section "We propose more precise versions of the Neural Collapse conjectures ("variability collapse"), stating
different versions for the train set and the test set, with both "strong" and "weak" forms." I only see `one` informal conjecture. Do you mean the `test` version of it? From the remark, "Refining our definitions and conjectures further is an area for future work," and its statement, this `conjecture` is far from precise.

The findings from the experiments in the "Train vs. Test Anti-Correlation" section require further clarification. The anti-correlation in Fig 4 is primarily attributed to the size of the training set, which can be explained without conducting the experiments. For instance, when each class has only one data point in the training set, the train variation is 0, but the test accuracy is close to that of random prediction. However, when the training set has a large number of samples, even if the train variation is high, the test accuracy is significantly higher. Therefore, it is unclear what additional insights the experiments in Figure 4 provide beyond this basic observation.


Discussion of implicit bias of SGD, maximal margin, and their relation to NC should be added. See, e.g., the line of work https://www.jmlr.org/papers/volume19/18-188/18-188.pdf. The NC phenomena can be viewed as the multi-class analog of the max-margin solution (also pure optimization phenomena) for binary classification.


**Strengths And Weaknesses:**

Detailed comments.

From theoretical prospect, the main claim from the paper "Neural Collapse is primarily an optimization phenomenon, and its connections to generalization require further investigation" is well recognized from the several immediate followup work of the original NC paper, e.g.  Lu & Steinerberger 2020, . Wojtowytsch et al. (2020) etc.

The main contribution from this paper, in my opinion, is to provide further empirical evidence to this claim. However, I also think experimental results are either known or trivial, partially owing to the lack of a clear causality relation between NC and generalization/better test features. That is to say one can find supporting `evidences` in either directions (NC can be good/bad for generalization etc.)

The formulation of Conjecture 2 is very confusing in my ways.  First, I expect a "formal" version of this "informal" conjecture, but I cannot find it in the context. Second, it can fail in many different ways, to name some, training blows up, networks are initialized in global minima that don't exhibit collapse.

Some conclusion is too strong. E.g. "Test Collapse implies Bad Representations", again, lacking causal relation between NC and representations, supporting evidences can be found either way. It may be better to soften that to "Test Collapse can cause Bad Representations"

---

### Review · Reviewer_qasE · 2023-02-14

**Summary Of Contributions:**

This paper empirically studies the limitations of the notion of "neural collapse" (specifically, NC1 from Papyan et al. (2020)) in understanding generalization of deep neural networks. In particular, it made separate definition of training and testing neural collapse and empirically measure them across a number of image classification datasets and neural network architectures. The paper found that while training NC does occur consistently, test NC in both strong and weak sense does not occur most of the time. Moreover, when varying the training set sizes, the train and test NC demonstrated an anti-correlation. The paper further show that test NC leads to worse performance in downstream task in transfer learning. As a result, this paper calls into question the value of NC in understanding deep neural network generalization.

**Audience:**

Yes

**Claims And Evidence:**

Yes

**Requested Changes:**

Please address the two main issues mentioned in the "weakness" section above, both of which are critical for my recommendation for acceptance.

**Strengths And Weaknesses:**

Strength

- This paper made precise definition and training and strong / weak test neural collapse
- This paper studies multiple datasets and architectures, and also studied in the context of transfer learning

Weakness

- While one of the main contribution of this paper stated in the introduction is "precise definition of train / test NCs", it seems there is no ambiguity in the original definition of neural collapse (specifically, NC1 as studied in this paper) from Papyan et al. (2020). Specifically, the definitions around page 4 and equation [3][4] made it very clear that the covariances were measured on the training set, therefore in this paper's term, measures the "train NC".

- The results on transfer learning is obvious and unrealistic. If NC happens on a pretraining task with very few classes, it cannot generalize well on downstream tasks with *more* classes. In more realistic setting, the pre-training datasets are usually much larger and contains a lot more classes than typical downstream tasks (e.g. pre-training on JFT and finetuning on Imagenet; or even pre-training on Imagenet and finetuning on CIFAR). Therefore, it would be far more interesting to study the "inverted" setting where the pre-training tasks have more classes (and data) than the downstream task.

---

### Public Comment · ~Avrajit_Ghosh1 · 2023-02-17
**Interesting work! Some queries and questions**

Interesting work by the authors! Could the authors shed more light on how (and if) learning rate (lr) affects either train collapse or "strong test collapse"? Since the paper studies neural collapse and generalization together, it may be interesting to see if higher lr (learning rate) makes test collapse more strong. I think the training collapse happens at convergence and should happen irrrespective of the learning rate (in stable regime). In SGD, larger lr is known to have higher test accuracy, I wonder whether the same trend would be observed as in Figure-4, i.e, the "strong test variance" decreases with larger lr (and better generalizaiton).

The authors seem to have fine-tuned the lr within the range  0.0005 to 0.2, could they comment on the above question then ?
        Also how "strong-test variance:" changes with batch-size (because it is an important factor in generalization as well) would be an important study (I see it is fixed at 128). All in all, it would be nice to have a figure that relates the "strong test variance" and the sharpness (l2 norm of Hessian), for various lr's and various batch-size for SGD.

---

### Decision · Action_Editors · 2023-03-08

**Recommendation:** Reject

**Comment:**

The lack of author participation in light of substantive concerns by the authors sealed the fate of the paper.

**Audience:**

The paper has a clear audience at TMLR

**Claims And Evidence:**

The reviewers are unanimous in stating that they believe the paper does not properly support its claims with convincing and clear evidence.

Given the lack of response from the authors, either to the comments or as changes to the manuscript to address the comments, I have to agree.